# Improving Brain Disorder Diagnosis with Advanced Brain Function Representation and Kolmogorov-Arnold Networks

**Tyler Ward**                                                    TYLER.WARD@UKY.EDU

**Abdullah-Al-Zubaer Imran**                                      AIMRAN@UKY.EDU

*University of Kentucky, Lexington, KY 40506, USA*

**Editors:** Accepted for publication at MIDL 2025

## Abstract

Quantifying functional connectivity (FC), a vital metric for the diagnosis of various brain disorders, traditionally relies on the use of a pre-defined brain atlas. However, using such atlases can lead to issues regarding selection bias and lack of regard for specificity. Addressing this, we propose a *novel* transformer-based classification network (ABFR-KAN) with effective brain function representation to aid in diagnosing autism spectrum disorder (ASD). ABFR-KAN leverages Kolmogorov-Arnold Network (KAN) blocks replacing traditional multi-layer perceptron (MLP) components. Thorough experimentation reveals the effectiveness of ABFR-KAN in improving the diagnosis of ASD under various configurations of the model architecture. Our code is available at `https://github.com/tbwa233/ABFR-KAN`.

**Keywords:** ASD, Brain MRI, classification, functional connectivity, Kolmogorov-Arnold Network, Transformer

## 1. Introduction

Diagnoses of autism spectrum disorder (ASD) are becoming increasingly prevalent across the world (Ge et al., 2024). As such, research into effective methods to improve the diagnosis of this brain disorder is vital. Traditional methods of diagnosing ASD have relied on the analysis of functional connectivity (FC) in the brain, quantified from blood-oxygen-level-dependent (BOLD) signals obtained during resting-state functional magnetic resonance imaging (rs-fMRI), but this approach has several flaws.

FC analysis performed in this matter typically relies on regions-of-interest (ROIs) produced by registering a subject's brain with a pre-defined atlas. This approach can lead to subjective selection bias, disregard for individual specificity, and a lack of interaction between brain regions and FC analysis (Liu et al., 2024a). Despite research into various methods of addressing these issues, such as data-driven (Jensen et al., 2024), individualized (Li et al., 2022), and multi-atlas (Xu et al., 2024) setups, a definitive resolution to all of the challenges associated with atlas-based parcellation techniques has not yet emerged.

Given that one of the largest drawbacks of traditional FC analysis is the high dimensionality and complexity of the functional representations, solutions that address this particular issue are desired. Recently, Kolmogorov-Arnold Networks (KANs) (Liu et al., 2024b) have emerged as an alternative to traditional multi-layer perceptrons (MLPs), leveraging learnable activation functions on edges rather than fixed activation functions on nodes. Inspired by the Kolmogorov-Arnold representation theorem, KANs replace conventional weight matrices with univariate functions parameterized as splines, offering improved expressiveness

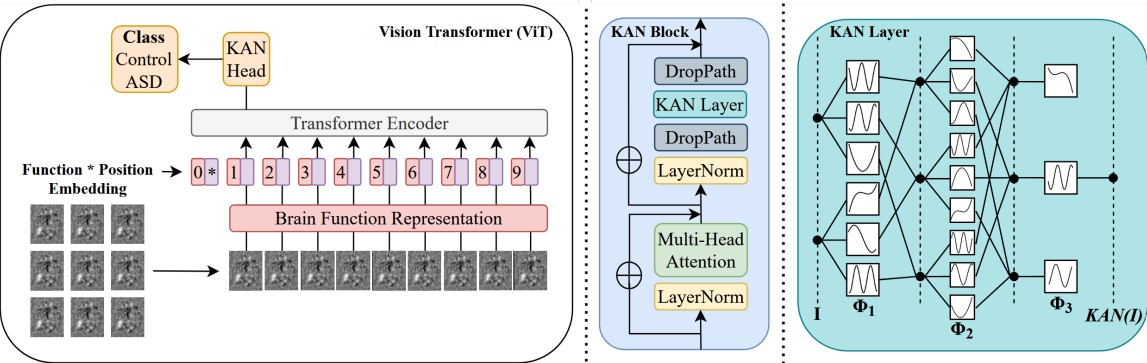

Figure 1: Proposed ABFR-KAN model: The transformer network is fed fMRI-derived patches that are embedded with spatial position information and passed through the encoder. The binary classification prediction (control or ASD) is produced by the KAN head. The encoder is in the ViT style, with a KAN block replacing the MLP block. The KAN block is similar to an MLP block but with DropPath regularization and KAN layers to handle nonlinear transformations. In the KAN layer, input $I$ passes through multiple learnable nonlinear functions ($\phi_n$) that are combined in a structured manner to form the final transformations. In our KAN implementation, a reflectional switch function is used as a basis function in the KAN layer.

and flexibility in function approximation. This design enables KANs to model complex transformations more efficiently while maintaining better interpretability and scaling properties compared to MLPs. Additionally, KANs have demonstrated potential in computer vision-related tasks (Cheon, 2024; Pal and Das, 2024). Based on this, we hypothesize that replacing MLPs with KANs in brain disorder diagnosis modes can better capture intricate relationships in FC patterns, leading to more robust and individualized ASD diagnoses.

In this paper, we propose *ABFR-KAN*, a novel workflow for brain disorder diagnosis. Building upon state-of-the-art methods, we propose novel sampling and function representation strategies and investigate the impact of KANs under various configurations in transformer networks. Our specific contributions are summarized as:

- Randomized anchor patch selection, which helps avoid structural bias, boosts individual-specific representations, and increases robustness and generalizability by reducing dependence on atlas-based parcellation.

- Iterative sampling of patches from a subject's brain, aimed to create multiple function representations for the same subject, introducing variance while preserving meaningful FC information.

- Extensive experimentation demonstrating the effectiveness of replacing traditional MLP components in two transformer networks (ViT and DeiT).

## 2. Related Work

There generally exist three different setups for brain disorder analysis using an atlas: single-atlas, multi-atlas, and individual-specific atlas. An example of a model constructed from a single-atlas approach is BrainGNN (Li et al., 2021), a graph neural network (GNN) based on the Desikan-Killiany (Desikan et al., 2006) atlas that is capable of analyzing fMRI images and discovering neurological biomarkers. Another group employed multiple atlases (Kennedy et al., 1998; Craddock et al., 2012; Rolls et al., 2020) to build a spectral GNN that enabled the identification of potential disease-related patterns associated with major depressive disorder (Lee et al., 2024). PFC-DBGNN- STAA (Cui et al., 2023) was proposed as a method for identifying mild cognitive impairment (MCI) based on individual-specific FC features.

As an alternative to using pre-defined atlases for ROI parcellation, several data-driven approaches have been proposed. For example, attention-guided hybrid deep learning networks have been used to localize discriminative brain regions automatically for Alzheimer's disease and MCI diagnosis (Lian et al., 2020). RandomFR (Liu et al., 2024a) is an innovative approach for brain function representation and operates via a randomized selection of brain patches as well as the use of novel function and position description methods. RandomFR serves as the main inspiration for the research presented in this paper.

Given the early stage of research into KANs, there exist few studies on the use of KANs for similar tasks that we propose in this paper. One study explores the use of KANs as deep feature extractors for MRI reconstruction, finding that incorporating Chebyshev polynomials into KANs (SS et al., 2024) led to both improved convergence and MRI reconstruction quality based on total variation and peak signal-to-noise ratio (Penkin and Krylov, 2024). Another study demonstrated the usefulness of KANs for chemical exchange saturation transfer (CEST) MRI analysis of the human brain (Wang et al., 2024), and another found that a model integrating the learnable spline activation functions of KAN into convolution layers, ConvKAN, outperformed traditional convolutional neural network (CNN) and graph convolution network (GCN) approaches at classifying Parkinson's disease (Patel et al., 2024). To our knowledge, we are the first to investigate the efficacy of KANs for FC analysis and ASD diagnosis.

## 3. Methods

In this work, we follow the workflow structure described by (Liu et al., 2024a) for brain function representation, which is divided into three stages: sampling, function representation, and transformer network. In the sampling stage, anchor patches are selected from the gray matter region of rs-fMRI scans. Each patch is defined by its average BOLD signal and its spatial position in the brain. In the function representation stage, sampled patches are characterized using a combination of function descriptions measuring FC and position descriptions, which encode their spatial locations in a standardized brain coordinate system. Function descriptions are computed as the Pearson correlation between the BOLD signal of the sampled patch and the anchor patches, forming a functional representation matrix. In the transformer network, embeddings based on the fusion of the function and position descriptions are passed to a transformer network for classification.

### 3.1. Random Anchor Selection

Previous works (Liu et al., 2024a) have explored selecting anchor patches using a grid-based method, where a grid of coordinates to sample anchor patches from is constructed from ROIs in a unified parcellation of gray matter, along with stride and offset values. This approach, which has proven effective, does limit models in terms of flexibility and adaptability because the same grid is used for every subject in a dataset, imposing a structural bias and a disregard for individual specificity, as a subject may have functionally distinct regions that do not align well with predefined anchor patches.

To address this flaw, we propose an alternate approach, a randomized one, for anchor patch selection. Our method works as follows. First, bounding boxes encompassing ROIs in the gray matter are calculated, and the starting coordinates are randomly sampled using:

$$x_{start} \sim \text{Uniform}(x_{min}, x_{max} - p_s), \tag{1}$$

where $p_s$ is the patch size. Similar equations are used to calculate $y_{start}$ and $z_{start}$. Once the anchor patches are sampled, they are validated to ensure sufficient overlap with the gray matter mask, $g_m$:

$$\sum(p_m \cdot g_m) \geq \tau, \tag{2}$$

where $p_m$ is the patch mask and $\tau$ is a threshold. If the condition is not met, resample until a valid sample is found. This process repeats until the desired number of anchor patches is sampled. A visualization of both the grid-based anchor selection and our random anchor selection can be seen in Figure 3.

### 3.2. Iterative Patch Sampling

Given an fMRI volume $V \in \mathbb{R}^{T \times X \times Y \times Z}$, where $T$ represents the number of timepoints and $X, Y, Z$ denote spatial dimensions, we define a grey matter mask $M \in \{0, 1\}^{X \times Y \times Z}$, where $M(i, j, l) = 1$ if the voxel belongs to grey matter. To extract representative features, we randomly sample a set of candidate patch centers $S = \{(x_k, y_k, z_k)\}_{k=1}^{N}$ from a uniform distribution within the fMRI volume.

Each selected center is used to define a cubic patch, $P_k$, of size $p$, extending symmetrically around the center along all three spatial dimensions. The patch is constrained within the grey matter by applying an element-wise mask with $M$, ensuring only voxels, $V_k$, belonging to grey matter are retained.

For each valid patch, we compute the mean fMRI signal across all timepoints:

$$\bar{V}_k = \frac{\sum_{i,j,l} V_k(i,j,l)}{\sum_{i,j,l} P_k(i,j,l) \cdot M(i,j,l)}. \tag{3}$$

Additionally, the patch's spatial position is normalized relative to the fMRI volume dimensions.

To establish functional connectivity relationships, we compute correlation coefficients between sampled patches and predefined anatomical anchor regions. Each anchor region $A_i$ is represented as a binary mask corresponding to a known brain region. The mean signal for each anchor is computed as:

$$\bar{V}_{A_i} = \frac{\sum_{i,j,l} V(i,j,l) \cdot \mathbb{1}[A(i,j,l) = i]}{\sum_{i,j,l} \mathbb{1}[A(i,j,l) = i]}. \tag{4}$$

Pairwise functional connectivity (FC) correlations between patch signals and anchor regions are computed using the Pearson correlation coefficient, $C_{ij} = \text{corr}(\bar{V}_i, \bar{V}_{A_j})$.

To capture fMRI features at varying spatial resolutions, the process is repeated across multiple patch sizes ($p \in \{8, 12, 16\}$). Each iteration samples new patches, extracts features, and computes FC matrices. The final aggregated FC matrix is obtained by averaging correlation values across iterations, and the final aggregated feature matrix consists of concatenated patch feature vectors across all iterations, capturing local brain activity features across multiple scales.

This iterative approach enhances the robustness of extracted fMRI features by reducing the impact of single-scale patch selection biases and improving anatomical coverage.

### 3.3. Transformer Network

In this work, we explore KAN integration in two popular transformer networks: vision transformer (ViT) (Dosovitskiy, 2020) and data-efficient image transformer (Touvron et al., 2021). A visual depiction of our implementation of the KAN-based ViT is shown in Figure 1. Both networks have two main locations that traditionally are constructed with MLP components: in the encoder and the classification head. We experiment with three different configurations of KAN integration: KAN-KAN, where the MLPs in both the encoder and classification head are replaced with KANs, KAN-MLP, where only the MLP in the encoder is replaced, and MLP-KAN, where the opposite is true.

## 4. Experiments and Results

### 4.1. Data

We evaluated our proposed ABFR-KAN using pre-processed neuroimaging data from the Autism Brain Imaging Data Exchange (ABIDE) (Craddock et al., 2013; Di Martino et al., 2014). The preprocessed ABIDE repository contains data collected from a total of 1,112 patients at various sites, preprocessed using a variety of methods. To initially train our ABFR-KAN model, we selected data from 171 patients that were collected from the New York University (NYU) Langone Medical Center site that had been processed using the Data Processing Assistant for Resting-State fMRI (DPARSF) (Yan and Zang, 2010) method. In total, 64 male (age range: 7-39) and 9 female (age range: 10-38) patients with ASD diagnoses were selected, along with 72 male (age range: 6-31) and 26 female (age range: 8-29) patients from the control group. To further validate the performance of ABFR-KAN on other ABIDE sites, as well as to explore its cross-domain performance, we select additional data from 110 patients collected from the University of Michigan (UM) Functional MRI Center. This time, the dataset was balanced, containing information from 55 patients in the control group (46 male aged 8 to 18, 9 female aged 9 to 18) and 55 in the ASD group (38 male aged 8 to 18, 17 female aged 9 to 19).

### 4.2. Implementation Details

The ABFR-KAN model was implemented with PyTorch and trained on on a *Intel (R) Xeon (R) w7-2475X, 2600MHz* machine with a dual *NVIDIA A4000X2* GPU (32GB). A 5-fold cross-validation strategy was used to assess the model's performance. For classification, we minimize the cross-entropy loss. The model was trained for 100 epochs, using the Adam optimizer with a learning rate of 0.0009. The model's performance is gauged using traditional metrics for classification tasks, namely accuracy (ACC), area under curve (AUC), F1 score (F1), precision (PRE), sensitivity (SEN), and specificity (SPE).

### 4.3. Results

From the results in Table 1, it is apparent that ABFR-KAN consistently outperforms MLP-only methods across different configurations, demonstrating the benefits of KANs in FC analysis. In the grid-based anchor selection, random patch sampling experiment, MLP-KAN performed best for ViT models, while KAN-KAN performed best for DeiT models. This suggests that KANs can enhance representation without fully replacing MLP components. It should be noted for this experiment that the MLP-MLP variant with the ViT backbone represents the RandomFR (Liu et al., 2024a) model.

For random anchor selection, random patch sampling, performance improved across most metrics, particularly for ViT models, indicating that removing structural bias allows for more subject-specific representations. KAN-KAN performed best for ViT models, while MLP-KAN was optimal for DeiT. The grid-based anchor selection, iterative patch sampling approach yielded more competitive results between ViT and DeiT, with KAN-MLP emerging as the top-performing configuration.

The random anchor selection, iterative patch sampling strategy achieved the highest classification performance, particularly with the ViT backbone and MLP-KAN configuration. This approach enhances robustness by introducing controlled variance while preserving subject specificity, effectively capturing individualized patterns in ASD diagnosis.

To assess generalizability, we evaluated the random anchor selection, random patch sampling approach on the ABIDE UM site (Table 2). Despite variations in data acquisition, ABFR-KAN demonstrated strong transferability, particularly for ViT-based models. Cross-domain evaluations, where models trained on one site (NYU) were tested on another (UM) and vice versa (Table 3), further confirmed its generalization capabilities.

Finally, we compared ABFR-KAN against traditional and state-of-the-art baselines (Table 4). Our method outperformed classical machine learning models like SVM and deep learning-based approaches such as BrainGNN and MVS-GCN. The most pronounced improvements were in sensitivity and AUC, highlighting ABFR-KAN's ability to improve ASD classification accuracy while reducing the misclassification of ASD-positive cases.

### 4.4. Discussion

Our results demonstrate the effectiveness of ABFR-KAN for FC-based ASD classification, with particular emphasis on the benefits of random anchor selection, iterative patch sampling. This configuration yielded the best performance across multiple metrics. In this section, we break down the contributing factors to this performance, analyze why the best-performing configuration (KAN-KAN) performed as it did.

Table 1: ABIDE NYU Site Performance: Classification performance of ABFR-KAN under different anchor selection and patch sampling strategies. The best and second best results are **bolded** and underlined, respectively.

| Backbone | Model | ACC | AUC | F1 | PRE | SEN | SPE |
|---|---|---|---|---|---|---|---|
| **Grid-based anchor selection, random patch sampling** | | | | | | | |
| ViT | MLP-MLP | 0.731±0.064 | 0.695±0.114 | 0.778±0.054 | 0.745±0.057 | 0.899±0.062 | 0.718±0.094 |
| | KAN-KAN | 0.708±0.092 | 0.706±0.080 | 0.768±0.039 | 0.762±0.145 | 0.867±0.121 | 0.692±0.124 |
| | KAN-MLP | 0.714±0.049 | **0.718±0.089** | 0.771±0.057 | **0.765±0.108** | 0.866±0.093 | 0.704±0.068 |
| | MLP-KAN | **0.737±0.050** | 0.686±0.096 | **0.785±0.050** | 0.741±0.070 | **0.910±0.058** | **0.719±0.049** |
| DeiT | MLP-MLP | 0.720±0.050 | 0.696±0.097 | 0.762±0.050 | 0.724±0.080 | 0.856±0.011 | 0.693±0.057 |
| | KAN-KAN | **0.725±0.057** | **0.697±0.049** | **0.773±0.043** | **0.750±0.131** | 0.846±0.060 | **0.711±0.068** |
| | KAN-MLP | 0.696±0.053 | 0.663±0.097 | 0.756±0.048 | 0.742±0.121 | 0.826±0.123 | 0.672±0.052 |
| | MLP-KAN | 0.679±0.058 | 0.662±0.025 | 0.767±0.017 | 0.733±0.080 | **0.939±0.049** | 0.663±0.066 |
| **Random anchor selection, random patch sampling** | | | | | | | |
| ViT | MLP-MLP | 0.702±0.033 | 0.667±0.081 | 0.737±0.057 | 0.723±0.061 | 0.776±0.125 | 0.704±0.039 |
| | KAN-KAN | **0.743±0.068** | **0.727±0.112** | **0.783±0.061** | **0.764±0.084** | **0.890±0.132** | **0.729±0.072** |
| | KAN-MLP | 0.720±0.045 | 0.704±0.041 | 0.768±0.055 | 0.734±0.082 | 0.825±0.116 | 0.699±0.037 |
| | MLP-KAN | 0.708±0.059 | 0.693±0.071 | 0.763±0.069 | 0.732±0.146 | 0.876±0.143 | 0.688±0.057 |
| DeiT | MLP-MLP | 0.696±0.072 | 0.671±0.092 | 0.755±0.065 | 0.725±0.080 | 0.865±0.139 | 0.682±0.085 |
| | KAN-KAN | 0.708±0.058 | 0.676±0.066 | 0.775±0.014 | 0.741±0.054 | 0.900±0.127 | 0.682±0.059 |
| | KAN-MLP | 0.684±0.051 | 0.622±0.020 | 0.767±0.032 | 0.722±0.117 | **0.959±0.050** | 0.665±0.065 |
| | MLP-KAN | **0.713±0.058** | **0.689±0.060** | **0.779±0.038** | **0.762±0.114** | 0.950±0.045 | **0.687±0.082** |
| **Grid-based anchor selection, iterative patch sampling** | | | | | | | |
| ViT | MLP-MLP | 0.643±0.047 | 0.612±0.040 | 0.760±0.049 | 0.725±0.043 | 0.839±0.124 | 0.652±0.040 |
| | KAN-KAN | 0.673±0.043 | 0.621±0.043 | 0.757±0.033 | 0.702±0.108 | 0.898±0.128 | 0.638±0.037 |
| | KAN-MLP | **0.690±0.048** | **0.674±0.090** | **0.772±0.035** | 0.683±0.045 | **0.918±0.068** | 0.651±0.051 |
| | MLP-KAN | 0.684±0.029 | 0.623±0.056 | 0.725±0.043 | **0.733±0.064** | 0.819±0.153 | **0.671±0.048** |
| DeiT | MLP-MLP | 0.667±0.047 | 0.608±0.089 | 0.733±0.029 | 0.685±0.063 | 0.844±0.156 | 0.647±0.055 |
| | KAN-KAN | 0.673±0.020 | 0.588±0.095 | 0.755±0.018 | 0.688±0.051 | 0.888±0.081 | 0.646±0.023 |
| | KAN-MLP | **0.707±0.054** | **0.667±0.101** | **0.779±0.030** | 0.706±0.075 | **0.898±0.085** | **0.674±0.064** |
| | MLP-KAN | 0.673±0.060 | 0.661±0.074 | 0.722±0.039 | **0.769±0.096** | 0.805±0.070 | 0.660±0.061 |
| **Random anchor selection, iterative patch sampling** | | | | | | | |
| ViT | MLP-MLP | 0.679±0.080 | 0.664±0.052 | 0.739±0.048 | 0.707±0.101 | 0.785±0.045 | 0.661±0.095 |
| | KAN-KAN | 0.703±0.082 | 0.669±0.094 | 0.764±0.075 | 0.698±0.062 | **0.919±0.084** | 0.678±0.079 |
| | KAN-MLP | 0.679±0.076 | 0.640±0.110 | 0.753±0.077 | 0.693±0.069 | 0.856±0.132 | 0.655±0.090 |
| | MLP-KAN | **0.743±0.088** | **0.734±0.131** | **0.786±0.069** | **0.780±0.151** | 0.897±0.065 | **0.716±0.105** |
| DeiT | MLP-MLP | 0.702±0.067 | **0.673±0.105** | 0.769±0.045 | **0.738±0.060** | 0.877±0.103 | **0.695±0.065** |
| | KAN-KAN | **0.714±0.047** | 0.645±0.045 | **0.781±0.020** | 0.707±0.059 | 0.909±0.062 | 0.683±0.064 |
| | KAN-MLP | 0.702±0.047 | 0.644±0.133 | 0.774±0.025 | 0.707±0.052 | 0.919±0.087 | 0.679±0.059 |
| | MLP-KAN | 0.690±0.058 | 0.645±0.102 | 0.763±0.042 | 0.687±0.061 | **0.928±0.051** | 0.658±0.070 |

Traditional grid-based anchor selection introduces structural biases by imposing predefined spatial constraints on the extracted FC patches. Our random anchor selection method mitigates this issue, allowing for more individualized functional representations. Iterative patch sampling provides a multi-scale view of each subject's FC patterns, introducing controlled variance while preserving meaningful functional signals. This is especially important

Table 2: ABIDE UM Site Performance: Classification performance of ABFR-KAN under the random anchor selection, random patch sampling setting. The best and second best results are **bolded** and underlined, respectively.

| Backbone | Model | ACC | AUC | F1 | PRE | SEN | SPE |
|---|---|---|---|---|---|---|---|
| ViT | MLP-MLP | 0.736±0.078 | **0.707±0.111** | 0.726±0.096 | 0.727±0.028 | 0.889±0.222 | **0.727±0.073** |
| | KAN-KAN | 0.727±0.050 | 0.673±0.107 | **0.766±0.044** | **0.767±0.137** | **0.913±0.085** | 0.723±0.066 |
| | KAN-MLP | 0.718±0.045 | 0.653±0.147 | 0.719±0.045 | 0.723±0.075 | 0.761±0.292 | 0.692±0.069 |
| | MLP-KAN | **0.736±0.053** | 0.667±0.056 | 0.740±0.072 | 0.722±0.171 | 0.793±0.163 | 0.716±0.021 |
| DeiT | MLP-MLP | 0.727±0.076 | **0.689±0.112** | 0.715±0.096 | 0.755±0.126 | 0.741±0.161 | 0.710±0.084 |
| | KAN-KAN | **0.727±0.070** | 0.658±0.043 | **0.742±0.107** | **0.762±0.145** | 0.836±0.199 | **0.711±0.026** |
| | KAN-MLP | 0.709±0.022 | 0.663±0.117 | 0.733±0.024 | 0.752±0.129 | **0.838±0.057** | 0.682±0.041 |
| | MLP-KAN | 0.700±0.068 | 0.687±0.127 | 0.721±0.068 | 0.675±0.074 | 0.805±0.155 | 0.678±0.095 |

Table 3: Cross-site generalizability of the proposed ABFR-KAN in ASD detection under the random anchor selection, random patch sampling setting. The best and second best results are **bolded** and underlined, respectively.

| Backbone | Model | Test: UM Site (Train: NYU Site) | | | Test: NYU Site (Train: UM Site) | | |
|---|---|---|---|---|---|---|---|
| | | ACC | AUC | F1 | ACC | AUC | F1 |
| ViT | MLP-MLP | 0.639±0.030 | 0.628±0.076 | 0.675±0.052 | **0.543±0.058** | **0.551±0.087** | 0.559±0.074 |
| | KAN-KAN | **0.676±0.062** | **0.687±0.106** | **0.713±0.056** | 0.536±0.037 | 0.525±0.083 | **0.590±0.034** |
| | KAN-MLP | 0.655±0.041 | 0.665±0.039 | 0.704±0.050 | 0.528±0.033 | 0.509±0.115 | 0.557±0.035 |
| | MLP-KAN | 0.644±0.054 | 0.655±0.067 | 0.701±0.063 | 0.538±0.039 | 0.520±0.044 | 0.574±0.056 |
| DeiT | MLP-MLP | 0.564±0.058 | 0.621±0.085 | 0.522±0.045 | 0.553±0.058 | **0.568±0.092** | 0.567±0.076 |
| | KAN-KAN | **0.578±0.047** | 0.623±0.061 | 0.536±0.010 | **0.559±0.054** | 0.546±0.036 | **0.586±0.085** |
| | KAN-MLP | 0.555±0.041 | 0.576±0.019 | 0.531±0.022 | 0.543±0.017 | 0.547±0.097 | 0.578±0.019 |
| | MLP-KAN | **0.578±0.047** | **0.636±0.055** | **0.542±0.026** | 0.534±0.052 | 0.564±0.104 | 0.575±0.054 |

in ASD classification, where inter-subject variability is high. The benefits of iterative patch sampling are evident in our best-performing configuration (MLP-KAN combined random anchor selection), as it enables the model to refine functional representations over multiple iterations.

While hybrid models demonstrated strong performance in some cases, notably in our best-performing model, the fully KAN-based configuration ultimately achieved the best results, as validated by a Kruskal-Wallis test followed by a Dunn test on the random anchor selection, iterative patch sampling experiment. These tests uncovered a statistically significant p-value of 0.0018 for DeiT models with the KAN-KAN configuration. We propose several explanations for this. First, the combination of KAN layers in both the encoder and classification head enables ABFR-KAN to model functional connectivity patterns with greater expressivity. KANs inherently capture complex, nonlinear relationships better than MLPs, making them particularly well-suited for FC analysis.

Additionally, KANs provide a more adaptable function representation, and having them in both the encoder and classification head ensures that the entire model benefits from this increased flexibility. While hybrid configurations (KAN-MLP, MLP-KAN) still offer improvements over MLP-only architectures, they lack the full expressive power of KAN-KAN, which may explain why they perform slightly worse in our experiments.

Table 4: Performance comparison of ABFR-KAN against the baseline and state-of-the-art models in ASD detection under the random anchor selection, random patch sampling setting. The best and second best results are **bolded** and underlined, respectively.

| Model | ACC | AUC | SEN | SPE |
|---|---|---|---|---|
| SVM (Cortes and Vapnik, 1995) | 0.649±0.056 | 0.663±0.097 | 0.838±0.152 | 0.619±0.045 |
| BrainNetCNN (Kawahara et al., 2017) | 0.696±0.039 | 0.654±0.072 | 0.828±0.121 | 0.676±0.065 |
| GAT (Veličković et al., 2017) | 0.661±0.041 | 0.640±0.073 | 0.817±0.068 | 0.636±0.046 |
| GCN (Qin et al., 2022) | 0.720±0.063 | 0.705±0.078 | 0.848±0.104 | 0.700±0.075 |
| BrainGNN (Li et al., 2021) | 0.719±0.030 | 0.663±0.048 | 0.784±0.108 | 0.706±0.032 |
| MVS-GCN (Wen et al., 2022) | 0.726±0.083 | 0.695±0.099 | 0.888±0.074 | 0.695±0.099 |
| ABFR-KAN | **0.743±0.088** | **0.734±0.131** | **0.897±0.065** | **0.716±0.105** |

Secondly, the number of trainable parameters varies across configurations, with KAN-KAN having the highest parameter count (190,112 for ViT and 205,072 for DeiT). While higher parameter counts can sometimes lead to overfitting, in this case, the added complexity appears to be beneficial. The learnable activation functions allow the model to adapt more flexibly to the underlying data structure rather than being constrained by fixed activation functions.

Although KAN-KAN was the best overall performer, our results also highlight scenarios where hybrid configurations offer advantages. Notably, MLP-KAN achieved a strong performance in grid-based anchor selection experiments, suggesting that conventional MLP heads may still be beneficial in certain cases. Such instances could be in settings where computational efficiency is a priority, where fewer trainable parameters would give a speed boost to the model.

## 5. Conclusions

In this paper, we have introduced a novel architecture for improving ASD diagnosis, ABFR-KAN. We systematically evaluated multiple architectural configurations, identifying the impact of KAN integration at different stages of the model pipeline. The most notable result was that the KAN-KAN configuration, which fully replaces MLP components with KAN layers, achieved the best classification performance, highlighting the expressive power of KANs in modeling nonlinear relationships in brain connectivity data. Additionally, we demonstrate that random anchor selection and iterative patch sampling provide substantial improvements in FC analysis by mitigating structural biases and introducing controlled variance.

Despite KAN-KAN's superiority, hybrid models such as MLP-KAN and KAN-MLP also exhibited strong performance in certain settings, indicating that selectively integrating KANs can provide a balance between expressive function representation and computational efficiency. These results suggest that while full KAN integration is optimal in many cases, hybrid configurations remain a viable alternative, particularly when computational constraints are a concern.

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

## Appendix A. Dataset Samples

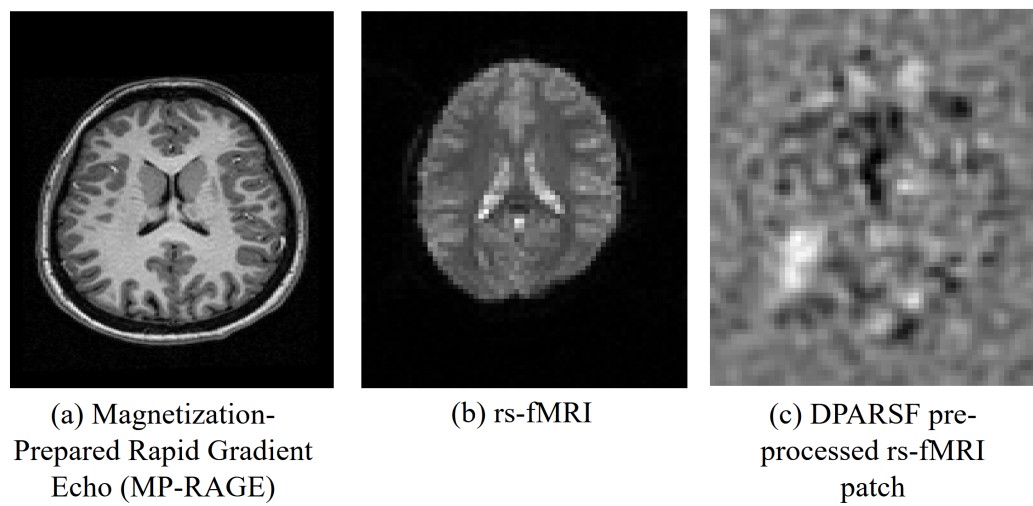

(a) Magnetization-Prepared Rapid Gradient Echo (MP-RAGE)

(b) rs-fMRI

(c) DPARSF pre-processed rs-fMRI patch

Figure 2: Raw and preprocessed data for a single patient in the ABIDE I dataset.

Figure 2 shows different patient-specific scans from the ABIDE I dataset. The first, MP-RAGE, is an MRI pulse sequence optimizing for T1-weighted imaging, allowing easy identification of anatomical features with high gray/white matter contrast (Brant-Zawadzki et al., 1992). The second, rs-fMRI, is a noninvasive technique used to measure and analyze brain activity when a subject is at rest, i.e., not engaged in a specific task. rs-fMRI's are widely used to study FC between brain regions (Santana et al., 2022). The third, and the most relevant to this research, is the DPARSF pre-processed fMRI data. DPARSF is a toolkit enabling easy pre-processing tasks such as slice timing, realignment, normalization, and smoothing data (Yan and Zang, 2010).

## Appendix B. Anchor Patch Selection

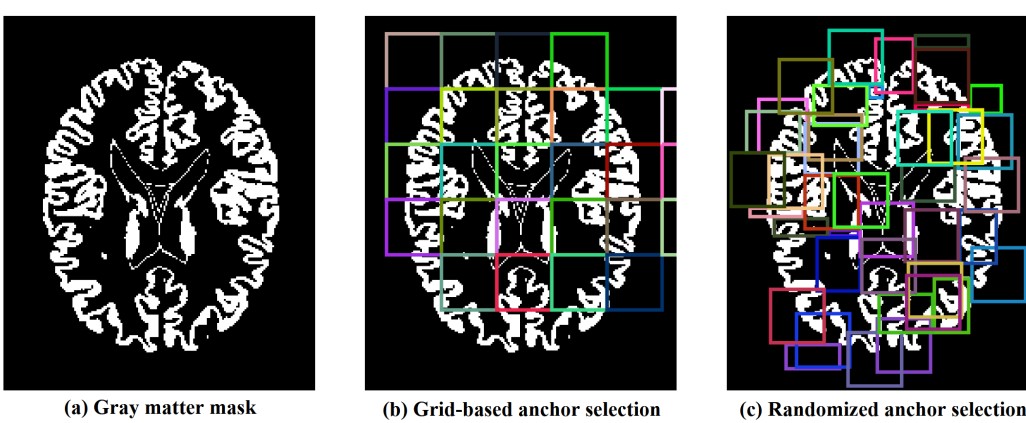



**(a) Gray matter mask**   **(b) Grid-based anchor selection**   **(c) Randomized anchor selection**



Figure 3: (a) The gray matter mask from which our anchor patches are selected. (b) The baseline grid-based anchor selection process. Note how certain patches fall outside of the gray matter region entirely. (c) Our randomized anchor selection process, which captures the full scope of the gray matter region, reduces structural bias and enhances individual specificity.

## Appendix C. Patch Sampling

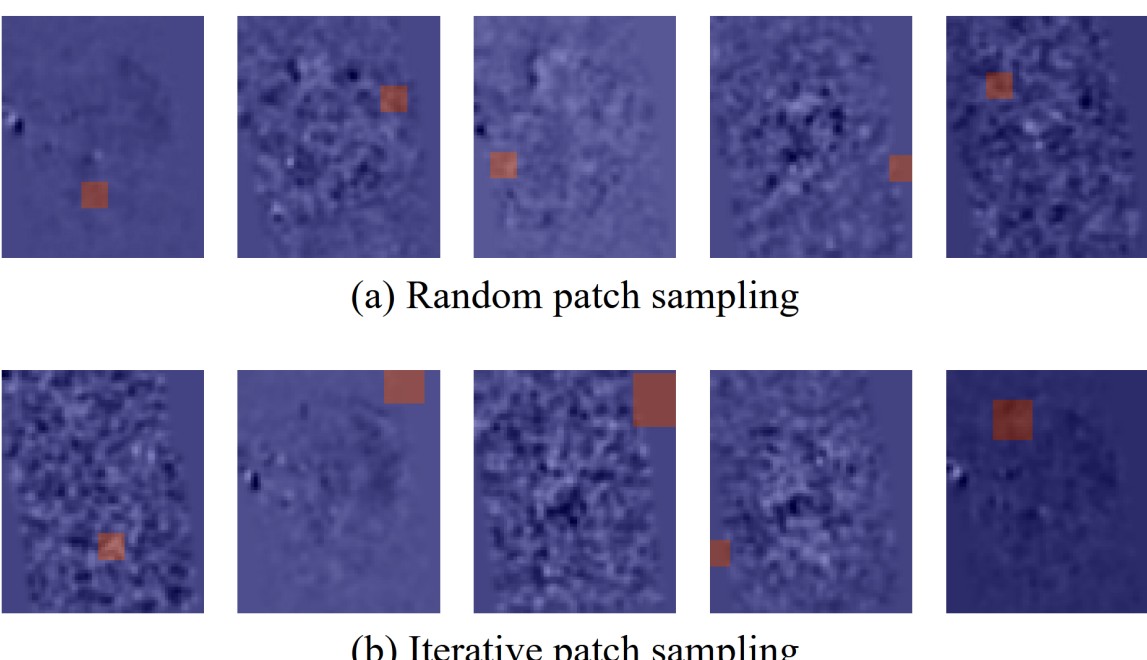

(a) Random patch sampling

(b) Iterative patch sampling

Figure 4: (a) The random patch sampling process. Observe how the size of the patches is consistent. (b) The iterative patch sampling process, where each subject is processed three times as a form of data augmentation, with patch sizes varying from 8×8, 12×12, and 16×16.

Figure 4 shows the two different patch sampling processes used in our study. The first, random patch sampling, uses consistent patch sizes randomly selected from across the gray matter region, reducing structural bias while maintaining functional specificity. The second, the iterative sampling method, acts as a data augmentation technique with the aim of introducing variability while preserving meaningful FC information.

## Appendix D. Receiver Operating Characteristic (ROC) Curves

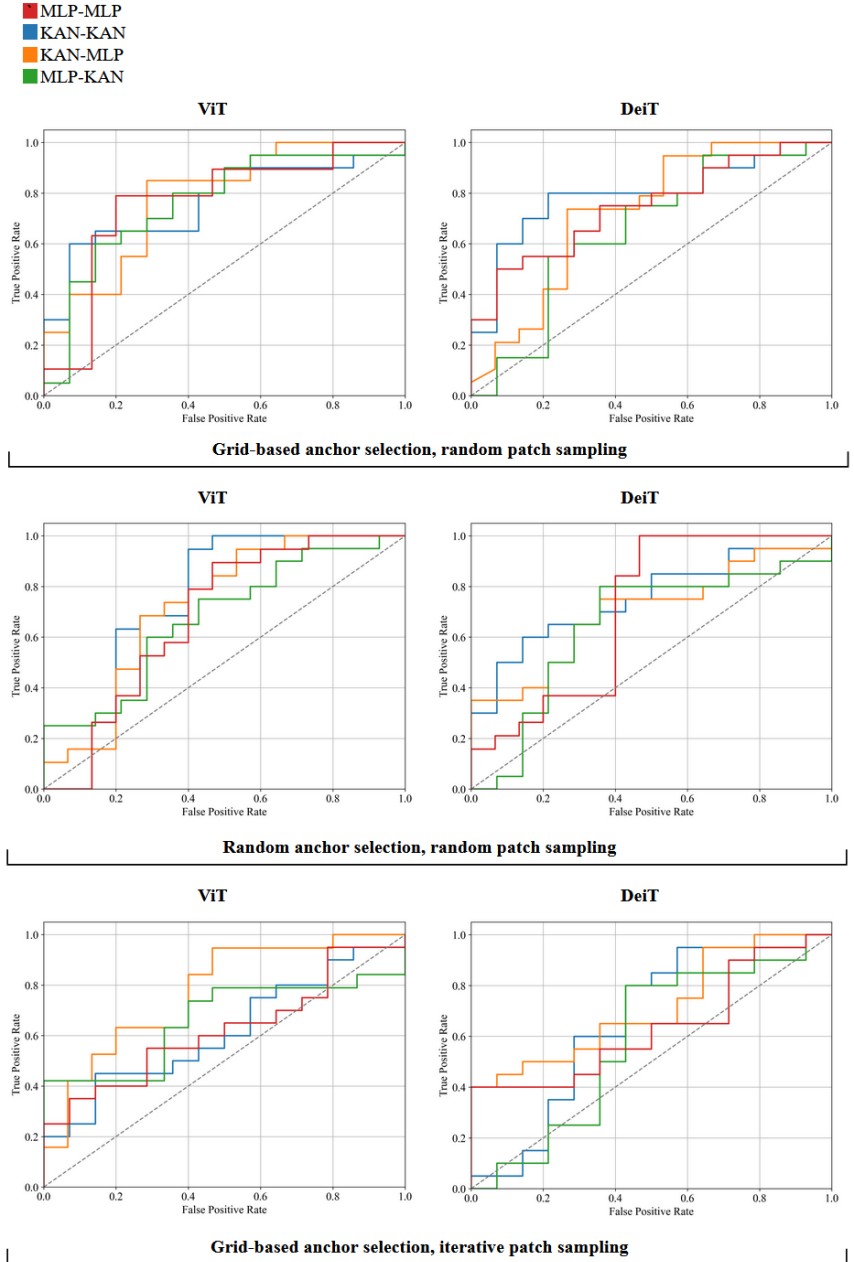

Figure 5: ROC curve comparison of our ABFR-KAN models vs. the baseline, which is reported as MLP-MLP. Note that our ABFR-KAN models generally achieve better curves compared to the MLP-MLP models.

Figure 5 presents a visual performance comparison between our ABFR-KAN models and the baseline. The ROC curves illustrate the trade-offs between true and false positive

rates. Our ABFR-KAN models demonstrate consistent performance improvement over the baseline, as indicated by the higher ROC curves. This suggests that replacing traditional MLP components with KANs enhances the model's ability to distinguish between ASD and control subjects. The observed improvements highlight the effectiveness of our approach in capturing complex FC patterns within the brain.

