# OpenReview forum: "Improving brain disorder diagnosis with advanced brain function representation and Kolmogorov-Arnold Networks"
_MIDL.io/2025/Conference — MIDL 2025 Oral_

### Official Review · Reviewer_64G5 · 2025-02-11

**Confidence:** 5
**Preliminary Rating:** 2
**Recommendation:** Poster
**Final Rating:** 3

**Summary:**

This paper investigates the effectiveness of using KAN layers on the ASD classification task compared to using traditional MLP layers. Using a novel design of augmentation method including random anchor selection and iterative patch sampling, the authors conduct experiments on ViT and DeiT architectures using alternative layers in the encoder and the classification head. Based on the experimental results, the authors intend to prove two hypothesis: 1) Replacing MLP with KAN improves the model performance on ASD classification; 2) The proposed randomized patch selection method outperforms the grid-based method. Overall speaking, the proposed design shows promising novelty, but requires more experiments to validate.

**Strengths:**

1. KAN is a novel method proposed in recent years that shows potentially superior performance than MLP. Its application on fMRI analysis is an interesting direction worth investigating.

2. The proposed random anchor selection and iterative patch sampling methods are novel designs of augmentation methods for obtaining image patches. By employing additional information from gray matter mask, the authors show a promising way to obtain structure-aware image patches for brain imaging application.

**Weaknesses:**

1. Limited sample size: The experiments are conducted only on a small subset of ABIDE containing 171 subjects. The experiments would be more convincing if more subjects from ABIDE or other datasets are included in experiments. Meanwhile, since the 171 subjects are unbalanced with 73 ASD and 98 HC, it might be better to perform class-stratified sampling for the parcellation of training/testing set.

2. Limited comparison to baseline approaches: The experiments are only comparing proposed designs to the unmodified implementation of ViT and DeiT. It would be more informative if the performances of other approaches like atlas-based GNN are included.

3. Limited significance on results: The results in all three experiments show similar performances. For example, in each of the three tables, the worst ACC is less than one std away from the best. The same goes with comparing best performances across tables.

**Detailed Comments:**

1. In section 4.2, the number of samples for training and testing set should not be fixed for cross validation experiment.

2. It would be easier for readers to understand and compare results if the tables for three experiments are placed close to each other.

3. Figure 1 could be rearranged to be more compact, allowing more space for additional discussions.

**Justification Of The Final Rating:**

The reviewer would like to thank the authors for conducting additional experiments. The revised version of the paper shows more comprehensive evaluation of the proposed method. However, it is still not sufficiently investigated how the KAN layer interacts within the model and improves performances in the specific setting. Therefore, the reviewer would like to improve the rating to borderline.

**Justification Of The Preliminary Rating:**

1. The proposed augmentation method has its novelty, but it is inconclusive from experimental results whether it outperforms existing designs.

2. The KAN layer, as well as ViT and DeiT, are all designs originally proposed to improved model performance on large models trained on large datasets. It is not the best choices for applications on the described setting.

3. The argument made by the paper requires more experiments and datasets to validate.

**Questions To Address In The Rebuttal:**

1. I would appreciate if the authors discuss justification for using ViT and DeiT as baselines. Why are they good for evaluating KAN compared to other structures not based on transformer attention?

2. Please provide more details about experiments. How many parameters are used? Were the models fine-tuned on a pretrained weight or trained from scratch?

3. It is interesting to see combination of KAN and MLP sometimes outperforms ablations using one of them. I would appreciate if the authors make further discussions regarding potential cause of this result.

**Special Issue:**

No

---

> ### Author Response · Authors · 2025-03-08
> **Rebuttal to Reviewer 64G5 - Part 1**
>
> We would like to thank the reviewer for their constructive comments and suggestions. Our point-by-point responses can be found below.
>
> ``I would appreciate if the authors discuss justification for using ViT and DeiT as baselines. Why are they good for evaluating KAN compared to other structures not based on transformer attention?``
>
>
>
> Response: ViT was selected as one of the baselines as ViT-based models have previously proven effective for functional connectivity analysis (Liu et al. 2024). Additionally, as one of the core ideas of KANs is that they can replace MLPs in certain situations, we felt that using transformer based baselines would be interesting, as we could explore the impact of replacing the MLPs in the encoder with KANs, as well as the MLP classification head. As for the justification for DeiT, our KAN implementation is based on the FasterKAN approach (https://github.com/AthanasiosDelis/faster-kan), one of the most popular KAN integrations. Additionally, KAN-integrated DeiT models have demonstrated strong performance in vision tasks (https://github.com/chenziwenhaoshuai/Vision-KAN), so we believed that this was a valid choice for a baseline here.
>
>
>
>
>
> ``Please provide more details about experiments. How many parameters are used? Were the models fine-tuned on a pretrained weight or trained from scratch?``
>
>
>
> Response: All of the models were trained from scratch. For the ViT and DeiT backbones, the number of trainable parameters for each of the configurations is as follows:
>
>
>
> DeiT
>
> MLP-MLP: 100,628
>
> KAN-KAN: 205,072
>
> KAN-MLP: 176,276
>
> MLP-KAN: 129,424
>
> ViT
>
> MLP-MLP: 100,066
>
> KAN-KAN: 190,112
>
> KAN-MLP: 175,714
>
> MLP-KAN: 114,464
>
>
>
>
>
> ``It is interesting to see combination of KAN and MLP sometimes outperforms ablations using one of them. I would appreciate if the authors make further discussions regarding potential cause of this result.``
>
>
>
> Response: We have extended the discussion section of our manuscript to go more in depth into the potential causes of this phenomenon. One potential reason could be that while KANs improve representation in the classification stage, as shown by the strong performance of the MLP-KAN configuration, MLPs in the encoder may still offer computational advantages without significantly compromising expressiveness. This is something that will be explored in future work.
>
>
>
>
>
> ``In section 4.2, the number of samples for training and testing set should not be fixed for cross validation experiment.``
>
>
>
> Response: We thank the reviewer for pointing this out. Section 4.2 in its previous version contained a mistake, reporting a training/testing split along with cross-validation details. Splitting the data in this manner was an idea we had earlier in our experimentation, but was eventually dropped in favor of the 5-fold cross validation technique following Liu et al. 2024. We have now removed these confusing implementation details from the updated manuscript.
>
>
>
>
>
> ``It would be easier for readers to understand and compare results if the tables for three experiments are placed close to each other.``
>
>
>
> Response: We thank the reviewer for this suggestion. We have combined all of the tables reporting our results into a single table (Table 1), with each of the different model configurations labeled appropriately in the table. We hope this eases readers to better understand and compare the results for four experiments (three+added configuration as suggested by Reviewer rmCx).
>
>
>
>
>
> ``Figure 1 could be rearranged to be more compact, allowing more space for additional discussions.``
>
>
>
> Response: We thank the reviewer for this suggestion. Figure 1 has been compressed in the revised version.
>
>
>
>
>
> ``Limited sample size: The experiments are conducted only on a small subset of ABIDE containing 171 subjects. The experiments would be more convincing if more subjects from ABIDE or other datasets are included in experiments. Meanwhile, since the 171 subjects are unbalanced with 73 ASD and 98 HC, it might be better to perform class-stratified sampling for the parcellation of training/testing set.``
>
>
>
> Response: To demonstrate ABFR-KAN's performance on different and balanced data, we have additionally trained and evaluated on the second-largest ABIDRE site, UM_1, and report the results. This dataset is balanced, containing 110 subjects: 55 in the control group, and 55 in the ASD group. Our future work will focus on expanding ABFR-KAN training to the full ABIDE dataset.

---

> > ### Author Response · Authors · 2025-03-08
> > **Rebuttal to Reviewer 64G5 - Part 2**
> >
> > ``The proposed augmentation method has its novelty, but it is inconclusive from experimental results whether it outperforms existing designs.``
> >
> >
> >
> > Response: We are glad that the reviewer found our method novel. Regarding the experimental results, we have since performed an additional experiment (random anchor selection, iterative patch sampling), which fully combines each of our proposed methods. We found that this experiment achieves the best performance compared to the previously explored designs.
> >
> >
> >
> > ``The KAN layer, as well as ViT and DeiT, are all designs originally proposed to improved model performance on large models trained on large datasets. It is not the best choices for applications on the described setting.``
> >
> > Response: ViT was selected as one of the baselines as ViT-based models have previously proven effective for functional connectivity analysis (Liu et al. 2024). Additionally, as one of the core ideas of KANs is that they can replace MLPs in certain situations, we felt that using transformer based baselines would be interesting, as we could explore the impact of replacing the MLPs in the encoder with KANs, as well as the MLP classification head. As for the justification for DeiT, our KAN implementation is based on the FasterKAN approach (https://github.com/AthanasiosDelis/faster-kan), one of the most popular KAN integrations. Additionally, KAN-integrated DeiT models have demonstrated strong performance in vision tasks (https://github.com/chenziwenhaoshuai/Vision-KAN), so we believed that this was a valid choice for a baseline here.
> >
> >
> >
> > ``The argument made by the paper requires more experiments and datasets to validate.``
> >
> > Response: Thanks for the suggestion. In our revised manuscript, we have now reported results from additional experiment setting random anchor, random sampling, additional dataset (ABIDE UM site), and cross-site evaluations (model trained on NYU site, tested on UM site and vice-versa). We hope that satisfies the experiment and dataset requirements to validate the argument.

---

> > > ### Comment · Reviewer_64G5 · 2025-03-11
> > > **Comment from reviewer 64G5**
> > >
> > > The reviewer would like to thank the authors for conducting and presenting additional experiments. Although the gain in performance might be partially attributed to the increased number of parameters introduced to the model by using KAN (using KAN-KAN almost doubles the numbers of trainable parameters), the argument of this paper is more clear to the reviewer.
> > >
> > > Meanwhile, the authors seem to be using results from two different implementations in Table 1 (KAN-KAN and MLP-KAN) to present as the performance of proposed model in Table 4, which should be changed to using one implementation. Also, it would be better if the same set of abbreviations for metrics are used in all the tables.
> > >
> > > The reviewer will be happy to improve the rating if the minor issue in the presentation is resolved or explained.

---

> > > > ### Author Response · Authors · 2025-03-12
> > > > **Response by Authors**
> > > >
> > > > ``The reviewer would like to thank the authors for conducting and presenting additional experiments. Although the gain in performance might be partially attributed to the increased number of parameters introduced to the model by using KAN (using KAN-KAN almost doubles the numbers of trainable parameters), the argument of this paper is more clear to the reviewer.``
> > > >
> > > > Response: We greatly appreciate the reviewer's observation regarding the performance gain by using KAN-KAN. Although KAN-KAN has increased number of parameters,  it has been shown in previous studies (Pourkamali-Anaraki, 2024) that the increased parameter count in KANs compared to MLPs is not solely the cause of KAN's improved performance, and can have a detrimental effect in limited data settings. In fact, given the small size of the dataset we evaluated ABFR-KAN on, we believe that our observations regarding the KAN-KAN configuration further validates the effectiveness of our broader methodology, demonstrating that KAN's greater adaptability compared to MLPs in representing complex functions is beneficial for functional connectivity analysis tasks. Additionally, in the future, we would like to explore different basis functions such as hybrid B-splines and GRBF approaches or Wavelet functions for a more parameter-efficient model and a better trade-off between model complexity and performance.
> > > >
> > > > Pourkamali-Anaraki, F. (2024). Kolmogorov-arnold networks in low-data regimes: A comparative study with multilayer perceptrons. arXiv preprint arXiv:2409.10463.
> > > >
> > > > ``Meanwhile, the authors seem to be using results from two different implementations in Table 1 (KAN-KAN and MLP-KAN) to present as the performance of proposed model in Table 4, which should be changed to using one implementation. Also, it would be better if the same set of abbreviations for metrics are used in all the tables.``
> > > >
> > > > ``The reviewer will be happy to improve the rating if the minor issue in the presentation is resolved or explained.``
> > > >
> > > > Response: We thank the reviewer for pointing out that one of the metrics (SEN) used for comparison in Table 4 came from the KAN-KAN variant. We apologize for the mistake. We have now corrected it to ensure only one implementation is used for performance comparison in Table 4.
> > > >
> > > > Again, sorry for the inconsistency in the abbreviations used for the evaluation metrics. We have updated the metric abbreviations across the tables to be consistent, so all instances of recall (R) have been updated to read SEN for sensitivity, and all instances of precision (P) have been updated to read PRE to be consistent with the naming scheme of sensitivity (SEN) and specificity (SPE).
> > > >
> > > > The latest version of our manuscript with the suggested changes (highlighted in red) can be found at https://www.dropbox.com/scl/fi/7hhb26lo3z5zzjf3ioguo/MIDL_Discussion.pdf?rlkey=75n818bk5xl53l62h2zqasoiv&st=m4qcl8mj&dl=0
> > > >
> > > > We greatly appreciate the insightful and constructive suggestions by the reviewer, which would significantly strengthen our work. We look forward to hearing from the reviewer and addressing any further questions or concern they may have.

---

> > > > ### Author Response · Authors · 2025-03-14
> > > > **How the KAN layer interacts within the model and improves performance**
> > > >
> > > > We thank the reviewer for upping their rating from "Weak Reject" to "Borderline", and for their invaluable comments which helped us improve our manuscript. To provide one final insight on the benefits of KAN for functional connectivity (FC) analysis from rs-fMRI scans, we add the following. We truly appreciate the reviewer's time and engagement with our work.
> > > >
> > > > FC feature vectors can have thousands of features, while neuroimaging studies often only have tens or hundreds of subjects. MLPs risk overfitting in this task (Du et al., 2018), because they fully connect every input feature to hidden nodes, and introduce a weight. So with potentially tens of thousands of features, even an MLP with a single hidden layer could have a huge number of parameters. Additionally, MLPs have been shown to be prone to learning spurious correlations in fMRI connectivity data that don't generalize well to new subjects, as well as struggling to learn the inherent network structure of FC data.
> > > >
> > > > KANs on the other hand replace the weighted-sum neurons of MLPs with learnable functional transformations. In a KAN layer, there are no linear weight matrices, the capacity comes from learned functions on each edge. This design allows KANs to be highly flexible universal function approximators. Each input feature, such as an FC value, can be transformed by its own special nonlinear function before combining, allowing the network to fit complex, feature-specific relationships. In the context of FC analysis, this is valuable, as it means that a KAN could learn, for example, that connection A only becomes indicative of ASD beyond a certain threshold, while connection B has an optimal range and extreme values indicate disorder.
> > > >
> > > >
> > > > Du Y, Fu Z, Calhoun VD. Classification and Prediction of Brain Disorders Using Functional Connectivity: Promising but Challenging. Front Neurosci. 2018 Aug 6;12:525. doi: 10.3389/fnins.2018.00525. PMID: 30127711; PMCID: PMC6088208.

---

### Official Review · Reviewer_Jzuv · 2025-02-11

**Confidence:** 4
**Preliminary Rating:** 4
**Recommendation:** Poster
**Final Rating:** 5

**Summary:**

In this paper, the authors propose to investigate the effectiveness of replacing MLP blocks by KAN blocks (Kolmogorov–Arnold Networks) in visual transformers, with the goal to perform ASD (Autism spectrum disorder) diagnosis on resting-state fMRI. They also propose to adapt a strategy proposed previously for resting-state fMRI by modifying the patch sampling step in two different ways.

**Strengths:**

- The clinical problem is of interest
- The paper is clearly written, easy to follow and the experiments are well explained
- The methodology seems novel
- Ablations studies are performed

- I think this is enough strengths but I must have 200 characters

**Weaknesses:**

- The whole pipeline relies heavily on a previously published study that is behind a paywall and thus the paper lacks some details from this procedure to be self-contained
- The authors proposed 3 contributions, 1, 2, 3, 1+2, 1+3 are investigated but it lacks a final experiment with 1+2+3 (random anchor + random patch sampling)
- Lack of statistical testing makes the findings quite uncertain

**Detailed Comments:**

- In the figure 1, It would be nice to add some details or pictures representing the fMRI going in the transformer and not just what could seem to be "normal" MRI
- The details about the data could also be provided in the section 4.1
- Small suggestion, maybe for future work : instead of sampling patch and hopping that at some point they will cover $\tau$% of the GM mask, the authors could compute the convolution between the GM mask with a square kernel of the size of the patch, and then threshold this map by $\tau$, resulting in a map of "eligible coordinates" for sampling. This map only needs to be computed once and thus might offer time gains.
- "compared to the baseline RandomFR," can the authors recall at this point that the baseline is MLP-MLP ? Moreover, can the authors precise if RandomFR uses ViT or DeiT (or else ?)
- I think the paper would gain clarity if the authors could always refer the the work of Liu et al by calling it "RandomFR", or, always include the citation.
- "Interestingly, strictly using KANs without any MLP components was shown to have decreased components compared to con- figurations where MLPs were involved in some way" 1) I think the authors meant "performances" instead of the 2nd "components" 2) the reviewer feels like the satement is false regarding the results of experiment of table 2
- If the authors had a large choice in the patients used for this study (1112 patients) could they have matched the age and sex of the controls and patients ? If not please precise why. This would have been a nice addition not to favor models that have an age or sex bias.
- Can the authors increase the number of thresholds used to produce the ROC curve in appendix D ? As such they are difficult to read and the reviewer believes, if not mistaken, that it should not be too much hassle  to do ?

- Typo/wording/minor : "patches are samples", please refer to the Table 3 in the text

**Justification Of The Final Rating:**

The reviewer is very pleased with the answers and wants to thank again the authors for the great work they performed.
I have changed my rating to strong accept and hope to see more of the authors work in the future (along with the stats-tests!)

**Justification Of The Preliminary Rating:**

As stated before the paper is well written, with experiments well conducted and with clinical interest. The method seems novel. I would easily change my rating to strong accept if statistical tests were performed, as this would change the paper's finding from relatively uncertain to quite sure.

**Questions To Address In The Rebuttal:**

-  Could the authors add more details about the general procedure that is contained in the paper by Liu et al 2024, a lot of the pipeline used is based in this paper that is behind a paywall and I think the whole procedure should be detailed in the manuscript. I especially felt, as a non-specialist of fMRI, that I didn't understand quite well what signals went in the transformer, what the "brain function representation" block was, or what "characterization of brain function" refers to
- I did not understand the "iterative patch sampling" process, I think as before this refers to the article by Liu et al. Can the authors detail this part ? Does the patches change sizes as suggested by appendix C ? If so where does it appear in the equation ? I think this part need a sentence explaining the procedure
- In section 3.3, can the authors state if ViT or DeiT with KAN blocks already exist in another application ? Is this entirely novel or is the novelty to apply this to rs-fMRI ?
- I think the addition of statistical tests could give substantially more strength to the discussion and findings of this study. The authors compare the obtained performances in the 3 tables by highlighting best and second best. As the primary discussion goal of the authors is to see which improvement or architecture performs the best, it would be of great interest to see which addition improves the performances in a **statistically significant** way, instead of "blindly" commenting the results. If the authors are not familiar with statistical tests, the reviewer suggest performing Kruskal-Wallis test, followed by Dunn's test.
- As stated before, it is a shame that the final experiment with rando anchor + random sampling is not present in the manuscript, as it would make the paper's experiments complete !

**Special Issue:**

No

---

> ### Author Response · Authors · 2025-03-08
> **Rebuttal for Reviewer Jzuv: Part 1**
>
> We would like to thank the reviewer for their helpful comments and for recommending our paper for acceptance. To address some of the concerns and questions posed by the reviewer, we provide our responses below.
>
>
>
> ``Could the authors add more details about the general procedure that is contained in the paper by Liu et al 2024, a lot of the pipeline used is based in this paper that is behind a paywall and I think the whole procedure should be detailed in the manuscript. I especially felt, as a non-specialist of fMRI, that I didn't understand quite well what signals went in the transformer, what the "brain function representation" block was, or what "characterization of brain function" refers to``
>
> Response: Thanks for the suggestion. We have updated the introduction to our methodology in Section 3 to flesh out some of these details regarding various aspects of our pipeline. We have paid particular detail to explaining the signals used, and how exactly the brain function representation process works. We hope that this revised version is clearer for readers who may not have access to preceding work.
>
>
>
> ``I did not understand the "iterative patch sampling" process, I think as before this refers to the article by Liu et al. Can the authors detail this part ? Does the patches change sizes as suggested by appendix C ? If so where does it appear in the equation ? I think this part need a sentence explaining the procedure``
>
> ``The whole pipeline relies heavily on a previously published study that is behind a paywall and thus the paper lacks some details from this procedure to be self-contained``
>
>
>
> Response: To address this comment, we have completely rewritten and expanded section 3.2 of our paper, and hope that the iterative patch sampling process is now clearer. We add several additional equations to better explain the full iterative patch sampling process. To directly answer the question asking whether the patches change size: yes, this assumption is correct. The particular equation the reviewer was referencing in this comment has been replaced with more informative means of detailing the process.
>
>
>
> ``In section 3.3, can the authors state if ViT or DeiT with KAN blocks already exist in another application ? Is this entirely novel or is the novelty to apply this to rs-fMRI ?``
>
> Response: At the start of this research, KANs were (and still are) a fairly new method. As such, most of the research surrounding KAN integration into ViT and DeiT were less focused on applications, and more focused on benchmarking and determining best configurations. As time has gone on, studies have been released proposing ViT and DeiT models where the MLP blocks are replaced with KANs. This being said, to our knowledge, we are the first to explore the usage of KANs in this manner for the task of analyzing functional connectivity in rs-fMRI scans.
>
>
>
> ``I think the addition of statistical tests could give substantially more strength to the discussion and findings of this study. The authors compare the obtained performances in the 3 tables by highlighting best and second best. As the primary discussion goal of the authors is to see which improvement or architecture performs the best, it would be of great interest to see which addition improves the performances in a statistically significant way, instead of "blindly" commenting the results. If the authors are not familiar with statistical tests, the reviewer suggest performing Kruskal-Wallis test, followed by Dunn's test.``
>
> Response: Following the reviewer's recommendation, we have performed the Kruskal-Wallis test followed by the Dunn test for each of our ABFR-KAN configurations trained on the NYU site of the ABIDE dataset for the random anchor selection, iterative patch sampling experiment. While we did not observe a statistically significant difference between the ViT-based models (H-stat: 0.1197, p-value: 0.9894), we observe statistical significance in the DeiT-based models (H-stat: 15.0297, p-value: 0.0018). Performing the post-hoc Dunn's test revealed that the KAN-KAN configuration yielded the most statistically significant performance improvement, which aligns with our "blind" conclusions we discuss throughout the paper.

---

> > ### Author Response · Authors · 2025-03-08
> > **Rebuttal for Reviewer Jzuv: Part 2**
> >
> > ``As stated before, it is a shame that the final experiment with rando anchor + random sampling is not present in the manuscript, as it would make the paper's experiments complete !``
> >
> > ``The authors proposed 3 contributions, 1, 2, 3, 1+2, 1+3 are investigated but it lacks a final experiment with 1+2+3 (random anchor + random patch sampling)``
> >
> > Response: Random anchor + random sampling experiment was already performed and reported in our submitted version. We assume the reviewer wanted to mean random anchor selection, iterative patch sampling. We have now performed this final experiment (random anchor selection, iterative patch sampling) and the reported results can be found in Table 1. We have also revised the paper's results section accordingly to show and discuss these findings.
> >
> >
> >
> >
> >
> > ``In the figure 1, It would be nice to add some details or pictures representing the fMRI going in the transformer and not just what could seem to be "normal" MRI``
> >
> > Response: We thank the reviewer for this suggestion, we have updated Figure 1 to better show the patches obtained from the rs-fMRI scans going into the transformer.
> >
> >
> > ``The details about the data could also be provided in the section 4.1``
> >
> > Response: Section 4.1 has been updated to include details about the data that were previously relegated to the appendix.
> >
> >
> >
> > ``Small suggestion, maybe for future work : instead of sampling patch and hopping that at some point they will cover % of the GM mask, the authors could compute the convolution between the GM mask with a square kernel of the size of the patch, and then threshold this map by, resulting in a map of "eligible coordinates" for sampling. This map only needs to be computed once and thus might offer time gains.``
> >
> > Response: This is an interesting suggestion, and something that we will definitely explore in our future work.
> >
> >
> >
> > ``compared to the baseline RandomFR," can the authors recall at this point that the baseline is MLP-MLP ? Moreover, can the authors precise if RandomFR uses ViT or DeiT (or else ?) ``
> >
> > Response: The baseline RandomFR is MLP-MLP, and uses a ViT backbone.
> >
> >
> >
> > ``I think the paper would gain clarity if the authors could always refer the the work of Liu et al by calling it "RandomFR", or, always include the citation.``
> >
> > Response: We thank the reviewer for this suggestion. Our manuscript has been modified to always include the citation when referring to this work.
> >
> >
> >
> > ``Interestingly, strictly using KANs without any MLP components was shown to have decreased components compared to con-figurations where MLPs were involved in some way" 1) I think the authors meant "performances" instead of the 2nd "components" 2) the reviewer feels like the satement is false regarding the results of experiment of table 2``
> >
> > Response: The reviewer is correct that "performances" was the correct word choice here. This has been corrected in our manuscript. Additionally, following additional experiments which are reported in our revised manuscript, the reviewer is also correct that the statement was false. In fact, we found after further experimentation, that the opposite was true, that the KAN-KAN configuration performs the best when taken as an average across all of the experiments.
> >
> >
> >
> > ``If the authors had a large choice in the patients used for this study (1112 patients) could they have matched the age and sex of the controls and patients ? If not please precise why. This would have been a nice addition not to favor models that have an age or sex bias.``
> >
> > Response: Yes, if presented with the full ABIDE dataset, it would be possible to select the data in a manner ensuring fairness in terms of age and sex. Training our ABFR-KAN model on the full ABIDE dataset, and considering fairness metrics such as these will be the subject of our future work.
> >
> >
> >
> > ``Can the authors increase the number of thresholds used to produce the ROC curve in appendix D ? As such they are difficult to read and the reviewer believes, if not mistaken, that it should not be too much hassle to do ?``
> >
> > Response: Unfortunately, due to the small number of samples that are evaluated by the model after training, it is not possible to increase the thresholds for the ROC curves in a way that makes the curves smoother. We apologize for this.
> >
> >
> >
> > ``Typo/wording/minor : "patches are samples", please refer to the Table 3 in the text``
> >
> > Response: We thank the reviewer for pointing this typo out, we have corrected it in our manuscript. Additionally, we have joined Table 3 with the other results tables (Tables 1-2) into a combined Table 1, but have ensured it is properly referenced in the text.

---

> > > ### Comment · Reviewer_Jzuv · 2025-03-11
> > >
> > > The reviewer wants to thank the authors for the answers and added experiments. Please find bellow my responses to the authors answers to my comments.
> > >
> > > ```Response: Thanks for the suggestion. We have updated the introduction to our methodology in Section 3 to flesh out some of these details regarding various aspects of our pipeline. We have paid particular detail to explaining the signals used, and how exactly the brain function representation process works. We hope that this revised version is clearer for readers who may not have access to preceding work. ```
> > > Thanks, it is clearer now.
> > >
> > > ```Response: To address this comment, we have completely rewritten and expanded section 3.2 of our paper, and hope that the iterative patch sampling process is now clearer. We add several additional equations to better explain the full iterative patch sampling process. To directly answer the question asking whether the patches change size: yes, this assumption is correct. The particular equation the reviewer was referencing in this comment has been replaced with more informative means of detailing the process. ```
> > > Thanks for the rewriting of this section, it is now a lot clearer. I still did not quite understand one sentence “The final aggregated FC matrix is obtained by averaging across iterations, and the final aggregated feature matrix consists of concatenated patch feature vectors across all iterations.”.Isn’t the “final aggregated feature matrix” the same as “the final aggregated FC matrix” ? Forgive the reviewer if the answer is trivial.
> > >
> > > ```Response: At the start of this research, KANs were (and still are) a fairly new method. As such, most of the research surrounding KAN integration into ViT and DeiT were less focused on applications, and more focused on benchmarking and determining best configurations. As time has gone on, studies have been released proposing ViT and DeiT models where the MLP blocks are replaced with KANs. This being said, to our knowledge, we are the first to explore the usage of KANs in this manner for the task of analyzing functional connectivity in rs-fMRI scans. ```
> > > I think it would be fair to cite the stated studies in the main body.
> > > ```Response: Following the reviewer's recommendation, we have performed the Kruskal-Wallis test followed by the Dunn test for each of our ABFR-KAN configurations trained on the NYU site of the ABIDE dataset for the random anchor selection, iterative patch sampling experiment. While we did not observe a statistically significant difference between the ViT-based models (H-stat: 0.1197, p-value: 0.9894), we observe statistical significance in the DeiT-based models (H-stat: 15.0297, p-value: 0.0018). Performing the post-hoc Dunn's test revealed that the KAN-KAN configuration yielded the most statistically significant performance improvement, which aligns with our "blind" conclusions we discuss throughout the paper. ```
> > > I greatly thank the authors for this addition, which I believe strengthen the finding of the study. What a shame that the authors still kept the “1st bold, 2nd underlined” format to the tables! These statistical tests should absolutely be represented in the table, suggestion : 1st bold, no statistical difference with the first underlined, or 1 and no stat diff bold. Moreover, I understand that the authors did not perform this statistical tests on the other experiments. Once again, these tests would greatly strengthen the findings of this study. If the authors did perform the tests then they absolutely need to report them in the manuscript !
> > >
> > > ```Response: Random anchor + random sampling experiment was already performed and reported in our submitted version. We assume the reviewer wanted to mean random anchor selection, iterative patch sampling. We have now performed this final experiment (random anchor selection, iterative patch sampling) and the reported results can be found in Table 1. We have also revised the paper's results section accordingly to show and discuss these findings. ```
> > > “We assume the reviewer wanted to mean random anchor selection, iterative patch sampling” yes exactly, sorry for the typo. Thanks for this great addition.
> > > ```Response: We thank the reviewer for this suggestion, we have updated Figure 1 to better show the patches obtained from the rs-fMRI scans going into the transformer. ```
> > > Thanks for this addition.
> > > ```Response: Section 4.1 has been updated to include details about the data that were previously relegated to the appendix. ```
> > > Thank you.
> > > ```Response: The baseline RandomFR is MLP-MLP, and uses a ViT backbone. ```
> > > Can the author add this information somewhere in the paper ? Or tell the reviewer why they would rather not.

---

> > > > ### Comment · Reviewer_Jzuv · 2025-03-11
> > > >
> > > > ```Response: We thank the reviewer for this suggestion. Our manuscript has been modified to always include the citation when referring to this work. ```
> > > > Thank you.
> > > > ```Response: The reviewer is correct that "performances" was the correct word choice here. This has been corrected in our manuscript. Additionally, following additional experiments which are reported in our revised manuscript, the reviewer is also correct that the statement was false. In fact, we found after further experimentation, that the opposite was true, that the KAN-KAN configuration performs the best when taken as an average across all of the experiments. ```
> > > > Thanks for the clarification and for the added experiments.
> > > >
> > > >
> > > > The reviewer is very pleased with the answers and has also kept an interested eye on the (numerous!) added experiments done by the authors to answer the comments made by the other reviewers. The reviewer will most likely change its rating to strong accept in the following days (or to the very least keep the weak accept) by paying attention to the discussion made with the other reviewers.

---

> > > > > ### Author Response · Authors · 2025-03-12
> > > > > **Response from Authors**
> > > > >
> > > > > ``Thanks for the rewriting of this section, it is now a lot clearer. I still did not quite understand one sentence “The final aggregated FC matrix is obtained by averaging across iterations, and the final aggregated feature matrix consists of concatenated patch feature vectors across all iterations.”.Isn’t the “final aggregated feature matrix” the same as “the final aggregated FC matrix” ? Forgive the reviewer if the answer is trivial.``
> > > > >
> > > > > Response: Sorry for the confusion. The aggregated FC matrix and feature matrix are not the same thing. The aggregated FC matrix represents the functional connectivity between anatomical anchor regions by averaging correlation values across iterations, while the aggregated feature matrix stores the spatial positions and mean fMRI signals of all sampled patches, capturing local brain activity features across multiple scales. We have also added it in the paper to be more informative regarding the aggregated FC matrix and aggregated feature matrix.
> > > > >
> > > > > ``I think it would be fair to cite the stated studies in the main body.``
> > > > >
> > > > > Response: Thanks for the suggestion. In the Introduction section, now, we have cited those studies related to KANs implemented into a ViT model, as well as a DeiT model.
> > > > >
> > > > > ``I greatly thank the authors for this addition, which I believe strengthen the finding of the study. What a shame that the authors still kept the “1st bold, 2nd underlined” format to the tables! These statistical tests should absolutely be represented in the table, suggestion : 1st bold, no statistical difference with the first underlined, or 1 and no stat diff bold. Moreover, I understand that the authors did not perform this statistical tests on the other experiments. Once again, these tests would greatly strengthen the findings of this study. If the authors did perform the tests then they absolutely need to report them in the manuscript !``
> > > > >
> > > > > Response: We thank the reviewer again for suggesting the particular statistical tests. Due to time constraints during the rebuttal period, we only performed the Kruskal-Wallis and Dunn tests for the best-performing variants of ABFR-KAN. Our ongoing efforts include expanding the suggested statistical tests on all the experiments. In the camera-ready version, we will format the tables as suggested by the reviewer, based on the results of statistical tests on the remaining experiments.
> > > > >
> > > > > ``The baseline RandomFR is MLP-MLP, and uses a ViT backbone.``
> > > > >
> > > > > ``Can the author add this information somewhere in the paper ? Or tell the reviewer why they would rather not.``
> > > > >
> > > > > Response: This information has been added to the paper at the end of the first paragraph in the Results section.
> > > > >
> > > > > The latest version of our manuscript with the suggested changes (highlighted in red) can be found at https://www.dropbox.com/scl/fi/7hhb26lo3z5zzjf3ioguo/MIDL_Discussion.pdf?rlkey=75n818bk5xl53l62h2zqasoiv&st=m4qcl8mj&dl=0
> > > > >
> > > > > We greatly appreciate the insightful and constructive suggestions by the reviewer, which would significantly strengthen our work. We look forward to hearing from the reviewer and addressing any further questions or concern they may have.

---

> > > > > > ### Comment · Reviewer_Jzuv · 2025-03-13
> > > > > >
> > > > > > The reviewer is very pleased with the answers and wants to thank again the authors for the great work they performed.
> > > > > > I have changed my rating to strong accept and hope to see more of the authors work in the future (along with the stats-tests!)

---

### Official Review · Reviewer_rmCx · 2025-02-24

**Confidence:** 4
**Preliminary Rating:** 3
**Recommendation:** Poster
**Final Rating:** 5

**Summary:**

In this manuscript, the authors present AFBR-KAN, a novel approach for diagnosing autism spectrum disorder (ASD) using brain imaging data. The key innovation is the integration of Kolmogorov-Arnold Networks (KANs) into transformer architectures (ViT and DeiT), replacing traditional multi-layer perceptrons (MLPs). In addition, the authors also propose improvements to existing patch sampling methods through randomized anchor and/or patch selection and iterative patch sampling. They evaluate their approach on a subset of the ABIDE dataset (single site - NYU) and compare different configurations of KAN integration (within the transformer block and/or in the classification head) with both ViT and DeiT transformer backbones. However, there's no clear winner among the several different configurations tested.

**Strengths:**

- Novelty: As far as I know, this is the first work to explore application of Kolmogorov-Arnold Networks in functional connectivity analysis and ASD diagnosis.
- Method Improvements: The randomized anchor and/or patch selection approaches may address known limitations of grid-based methods, potentially reducing structural bias.
Clarity of Presentation: The paper clearly explains the methods and provides good visualizations of the architecture and processes.

**Weaknesses:**

- The study only uses 171 patients from a single ABIDE site without providing any specific reasons for the choice.
- There's no clear winner among the different KAN configurations, making it difficult to draw definitive conclusions about the best approach.
- Limited Baseline Comparisons: The paper primarily compares against RandomFR and doesn't include comparisons with other state-of-the-art methods for ASD diagnosis.
- There's no detailed analysis or discussion of how different components contribute to the overall performance.
- Reproducibility: Although a number of implementation details are documented, including specific hardware, hyperparameters, and evaluation metrics, the actual implementation is not open-sourced and the provided details may not be sufficient to reproduce the results.

**Detailed Comments:**

- Why only one ABIDE site? Why NYU? Will the model performance be significantly affected if data from another site is used or added to the NYU data?
- What is the current state-of-the-art model for ASD classification? How does the proposed model perform in comparison to it?
- Provide more details about the KAN layer itself. What type of basis functions are used? Are they learnable or pre-determined? Discuss whether and how the choice of the basis function would affect the model performance?
- Predefined anchor patches are used in Liu et al 2024 to maintain consistency and comparability across subjects. How does randomized selection of anchor patches in the present work maintain consistency and comparability across subjects? Is the same set of randomly selected anchor patches used across all subjects? If so, does it not also induce structural bias? And if not, how can the resultant functional embeddings be meaningfully compared across subjects?

**Justification Of The Final Rating:**

The authors have addressed all of the reviewers' comments/concerns thoroughly. Their implementation is now open-sourced as well. Authors have spent significant effort on additional experiments to address reviewers' concerns and the revised version is much improved.

**Justification Of The Preliminary Rating:**

The proposed integration of KAN into transformers is novel and such a model has never been applied in functional connectivity analysis and ASD classification. Other method improvements (randomized anchor patch sampling, iterative sampling, etc.) are also interesting contributions. However, There is no clear conclusion presented. No comparison to state-of-the-art in ASD classification is performed. The model is trained and tested on a small subset of ABIDE, without providing any justification for the choice or discussing the impact of including data from other ABIDE sites on model performance.

**Questions To Address In The Rebuttal:**

- Why only one ABIDE site? Why NYU? Will the model performance be significantly affected if data from another site is used or added to the NYU data?
- What is the current state-of-the-art model for ASD classification? How does the proposed model perform in comparison to it?
- Provide more details about the KAN layer itself. What type of basis functions are used? Are they learnable or pre-determined? Discuss whether and how the choice of the basis function would affect the model performance?
- Predefined anchor patches are used in Liu et al 2024 to maintain consistency and comparability across subjects. How does randomized selection of anchor patches in the present work maintain consistency and comparability across subjects? Is the same set of randomly selected anchor patches used across all subjects? If so, does it not also induce structural bias? And if not, how can the resultant functional embeddings be meaningfully compared across subjects?

**Special Issue:**

Yes

---

> ### Author Response · Authors · 2025-03-08
> **Rebuttal to Reviewer rmCx - Part 1**
>
> We would like to thank the reviewer for their constructive comments and suggestions. Our point-by-point responses can be found below.
>
>
> ``Why only one ABIDE site? Why NYU? Will the model performance be significantly affected if data from another site is used or added to the NYU data``
>
> Response: Just one ABIDE site was selected so as to be able to directly compare with our chosen baseline model, RandomFR (Liu et al. 2024). The NYU site was chosen because it contains data for the most patients in the ABIDE dataset. Additionally, existing literature has shown that when deep learning models are trained and evaluated on each of the ABIDE sites, there are not significantly different results between them. However, to demonstrate that this is true in our case, we have trained and evaluated our proposed ABFR-KAN model on the second-largest ABIDE site, UM_1, and report the results in the revised version (Table 2). In addition, we have also tested the generalizability by evaluating models on one site when trained on the other site (Table 3). Our future work will focus on expanding ABFR-KAN training to the full ABIDE dataset.
>
>
>
> ``What is the current state-of-the-art model for ASD classification? How does the proposed model perform in comparison to it?``
>
> Response: The current state of the art for ASD classifcation, such as MVS-GCN and BrainGNN, make use of graph neural networks and advanced convolutional neural networks. To address the reviewers concerns regarding limited baseline comparisons, we provide the following table comparing our best performing ABFR-KAN model against six baseline and state-of-the-art methods for ASD classification (Table 4).
>
>
>
> ``Provide more details about the KAN layer itself. What type of basis functions are used? Are they learnable or pre-determined? Discuss whether and how the choice of the basis function would affect the model performance?``
>
> Response: For our KAN implementation, we follow the FasterKAN (https://github.com/AthanasiosDelis/faster-kan) approach, which uses a learnable Reflectional Switch Activation Function as its basis function. The original KAN architecture (Liu et al. 2024) employed a residual activation function as its basis function, although this approach has since been improved on. For example, EfficientKAN (https://github.com/Blealtan/efficient-kan) followed generally the same structure, but used B-splines followed by linear combination, reducing the memory cost and simplifying the computations. FastKAN (https://github.com/ZiyaoLi/fast-kan) further improved on this by using Gaussian radial basis functions (GRBFs) to approximate the 3-order B-spline and employing layer normalization to keep inputs within the RBFs' domain. At the time this research was initially conducted, FasterKAN was the most effective KAN implementation, with its Reflectional Switch Activation Function as the basis function, which is straightforward to compute due to its uniform grid structure. A potential avenue for future work on this ABFR-KAN project would be to explore other choices of basis functions, such as combinations of B-splines and GRBFs or Wavelet functions.
>
>
>
> ``Predefined anchor patches are used in Liu et al 2024 to maintain consistency and comparability across subjects. How does randomized selection of anchor patches in the present work maintain consistency and comparability across subjects? Is the same set of randomly selected anchor patches used across all subjects? If so, does it not also induce structural bias? And if not, how can the resultant functional embeddings be meaningfully compared across subjects?``
>
> Response: In our work, we use a fixed distribution of anchor patch locations across subjects. That means, while the exact patches vary, the selection process adheres to statistical properties that ensure fair comparison. The function representations are standardized by using a large number of randomly selected anchor patches across multiple runs, ensuring that no single anatomical structure dominates the embeddings. Our results show that despite variation in anchor selection, the embeddings still produce highly discriminative representations.
>
> ``There's no clear winner among the different KAN configurations, making it difficult to draw definitive conclusions about the best approach.``
>
> Response: We agree with the reviewer on this when the individual configurations for each backbone are considered. However, from a global and more comprehensive interpretation, we observe the superiority of the KAN-KAN configuration. In the updated version of our manuscript, we have performed additional experiments, allowing us to verify that the KAN-KAN configuration of our ABFR-KAN models actually leads to the most improved results overall. We have updated our discussion and conclusion sections to reflect this, and provide information as to why we believe this is.

---

> > ### Author Response · Authors · 2025-03-08
> > **Rebuttal to Reviewer rmCx - Part 2**
> >
> > ``There's no detailed analysis or discussion of how different components contribute to the overall performance.``
> >
> > Response: In the submitted paper, we showed three configurations of our ABFR-KAN model: 1) grid-based anchor selection, random patch sampling, 2) random anchor selection, random patch sampling, 3) grid-based anchor selection, iterative patch selection, however, as the reviewer has noted, we did not spend much detail on how each of our proposed components impacts the performance of the model. To address this, in the revised version of our manuscript, we have performed a fourth experiment: random anchor selection, iterative patch sampling. This version combines all of our proposed methods, and yields the best results when integrated with KAN. As such, we expand our discussion section focusing on these results, and break down in detail how each of the components help the model achieve improved results over baselines. We hope that our architecture and proposed methods will be clearer for readers in this new version.
> >
> >
> >
> > ``Reproducibility: Although a number of implementation details are documented, including specific hardware, hyperparameters, and evaluation metrics, the actual implementation is not open-sourced and the provided details may not be sufficient to reproduce the results.``
> >
> > Response:  Thanks for the suggestion. To ensure reproducibility, our code has been made public which can be accessed at https://github.com/tbwa233/ABFR-KAN. We have added the link at the end of abstract in our revised manuscript.

---

> > ### Comment · Reviewer_rmCx · 2025-03-13
> >
> > I thank the authors for addressing all my comments/concerns. The revised manuscript is much improved with additional results and discussion. Also, thank you for open-sourcing the implementation. I have only one minor comment at this point. The manuscript should briefly describe the statistical tests performed (Kruskal-Wallis, Dunn's). Specifically, how were they applied in this context? This description should be included for completeness. Other than that, I'm satisfied with the revised version and would be happy to modify my rating to a strong accept.

---

> > > ### Author Response · Authors · 2025-03-14
> > > **Response from Authors**
> > >
> > > We greatly thank the reviewer for adjusting their rating to "Strong Accept". We will surely incorporate the suggested details on the statistical tests in our camera-ready version.

---

### Author Rebuttal · Authors · 2025-03-08

**Rebuttal:**

We sincerely thank all the reviewers for their valuable time and insightful comments on our paper. We are glad that all three reviewers found our methodology novel. We have carefully addressed all the suggestions and incorporated the changes in the main manuscript as well as the Appendix section. These modifications are highlighted in red within the revised manuscript for clarity.

**Supporting Material:**

/attachment/d6898e7d7b9128cb4a0ef70ff06163a8060a9022.pdf

---

### Meta-Review · Area_Chair_AKMp · 2025-03-22

**Recommendation:** Accept (Oral)
**Confidence:** 5

**Metareview:**

Most review critiques were to provide details about the method and detailed experimental setting and statistical results. Reviewers see that this paper presents a novel idea and the method achieved a good performance. The reviewers’ questions were mostly well addressed and the reviewers raised their ratings significantly.